# Utilization of Primary Healthcare Services in Patients with Multimorbidity According to Their Risk Level by Adjusted Morbidity Groups: A Cross-Sectional Study in Chamartín District (Madrid)

**DOI:** 10.3390/healthcare12020270

**Published:** 2024-01-20

**Authors:** Jaime Barrio-Cortes, Almudena Castaño-Reguillo, Beatriz Benito-Sánchez, María Teresa Beca-Martínez, Cayetana Ruiz-Zaldibar

**Affiliations:** 1Foundation for Biosanitary Research and Innovation in Primary Care (FIIBAP), 28003 Madrid, Spain; 2Faculty of Health, Camilo José Cela University, 28692 Madrid, Spain; 3Ciudad Jardín Health Center, Primary Care Management, 28002 Madrid, Spain

**Keywords:** multimorbidity, primary care, healthcare services utilization, patient management, morbidity grouper, stratification

## Abstract

Patients with multimorbidity have increased and more complex healthcare needs, posing their management a challenge for healthcare systems. This study aimed to describe their primary healthcare utilization and associated factors. A population-based cross-sectional study was conducted in a Spanish basic healthcare area including all patients with chronic conditions, differentiating between having multimorbidity or not. Sociodemographic, functional, clinical and service utilization variables were analyzed, stratifying the multimorbid population by the Adjusted Morbidity Groups (AMG) risk level, sex and age. A total of 6036 patients had multimorbidity, 64.2% being low risk, 28.5% medium risk and 7.3% high risk. Their mean age was 64.1 years and 63.5% were women, having on average 3.5 chronic diseases, and 25.3% were polymedicated. Their mean primary care contacts/year was 14.9 (7.8 with family doctors and 4.4 with nurses). Factors associated with primary care utilization were age (B-coefficient [BC] = 1.15;95% Confidence Interval [CI] = 0.30–2.01), female sex (BC = 1.04; CI = 0.30–1.78), having a caregiver (BC = 8.70; CI = 6.72–10.69), complexity (B-coefficient = 0.46; CI = 0.38–0.55), high-risk (B-coefficient = 2.29; CI = 1.26–3.32), numerous chronic diseases (B-coefficient = 1.20; CI = 0.37–2.04) and polypharmacy (B-coefficient = 5.05; CI = 4.00–6.10). This study provides valuable data on the application of AMG in multimorbid patients, revealing their healthcare utilization and the need for a patient-centered approach by primary care professionals. These results could guide in improving coordination among professionals, optimizing multimorbidity management and reducing costs derived from their extensive healthcare utilization.

## 1. Introduction

Multimorbidity is commonly defined as the presence of two or more chronic conditions in an individual [1,2,3]. There is no homogeneous universal definition for multimorbidity since it depends on the number, type and duration of the chronic diseases considered, as well as on the population and area studied and the data sources and data collection methods, among other factors [4]. 

The prevalence of chronic diseases has rapidly risen in the past years and is expected to continue increasing, and thus multimorbidity too. The main reasons are population aging and improvement in survival [5]. However, multimorbidity involves progressive clinical deterioration, increased disability, decline in quality and life expectancy, polypharmacy and increased health service utilization [1,2], especially in primary care [3].

Nowadays, multimorbidity poses a challenge for the management of patients with multimorbidity, as healthcare services usually focus on a single disease [3], leading to care that is sometimes inadequate and potentially harmful [1]. Therefore, patients with multimorbidity require novel multiple disease-specific strategies involving comprehensive and multidisciplinary patient-centered care, with disease-specific goals, together with careful prescription of multiple medications [1,2,3]. For designing a meaningful patient-centered approach, it is indispensable to characterize and understand the sequence of diseases, functional limitations and health service utilization of the population with multimorbidity, especially focusing on high-risk patients, in order to guide effective action for the improvement of clinical care and policy of patients with multimorbidity [6]. Nevertheless, there is still limited evidence to support any approach, so more research on multimorbidity is urgently needed [1].

The WHO has highlighted the fundamental role of primary care in the management of patients with multimorbidity [3,7]. To guide this management, many countries use morbidity groupers to stratify populations according to complexity [8]. In Spain, the Adjusted Morbidity Groups (AMGs) have been developed within the Spanish healthcare system and are integrated into the electronic medical records of primary care [9]. The AMG grouper enables the measurement of multimorbidity to determine its impact on clinical-care management, epidemiology and healthcare administration, while also classifying patients into risk categories based on their morbidity and complexity [10]. This tool is useful for primary care professionals and policymakers, as it reveals the characteristics and use of services in patients with multimorbidity, serving as a guide to allocate healthcare resources efficiently and to plan appropriate care models and interventions based on each individual risk level, thereby successfully meeting the healthcare needs of patients with multimorbidity and efficiently managing healthcare services [8,9,10].

As AMG is a relatively recent and useful tool, and as more research is needed to better characterize populations with multimorbidity for designing optimized care strategies, we aimed to describe the chronic conditions and the use of primary care services and associated factors in patients with multimorbidity according to their AMG risk level, sex and age, and comparing to those patients without multimorbidity.

## 2. Materials and Methods

### 2.1. Design, Setting and Study Subjects

A cross-sectional observational study was performed in a basic healthcare area located in the Chamartín district in Madrid (Spain). Chamartín is located in the north of Madrid city center and has 143,424 inhabitants, with an average age of 45 years (23% over 65 years), 55% women, 8.9% foreigners and corresponding to the lowest degree of socioeconomic deprivation. Madrid’s healthcare system follows the Spanish National Health System, which guarantees almost universal coverage for all residents, organizing the healthcare management of the population by dividing the country into basic healthcare areas staffed by primary care teams under the gatekeeping model [11].

All the patients with chronic conditions were included in the study, differentiating between those without and with multimorbidity. The healthcare area studied covered 18,107 people, of whom 9886 had chronic illnesses as of 30 June 2015.

### 2.2. Data Collection

Data were requested from the Information Health Systems Department of Madrid Primary Care, which extracted all the study information registered in the Madrid primary care electronic medical record database for each chronic patient. Chronic patients were identified via the AMG tool in the electronic clinical record of the Madrid primary care system, which considers as chronic patients those who present at least one chronic disease listed in Appendix A [7]. 

### 2.3. Variables

AMG classifies populations considering morbidity and complexity. The AMG algorithm assigns a relative weight to mortality risk, admissions, primary care visits and prescriptions for the diagnostic code grouping present in each patient, allowing the calculation of an individual level of complexity depending on morbidity. The complexity numerical index obtained enables the stratification of the population into four risk levels following the Kaiser-Permanente pyramid, which assigns cut-off points from the 50th, 80th and 95th percentiles of the population, 50% of which corresponds to the to the population of people without any relevant chronic pathology, 30% to patients with chronic diseases and with low risk levels, 15% to those with medium risk, and 5% to those with high risk [9,10].

Sociodemographic variables including age, sex and country of origin (Spain, Europe or the rest of the world) were determined. Also, functional status was characterized by the presence of immobilization at home, institutionalization in a nursing home, requiring a primary caregiver, having home support and being in palliative care. Clinical characteristics were identified by AMG risk level (into low, medium or high risk), AMG complexity index (numerical value of patient complexity assigned by AMG), number and type of chronic diseases, multimorbidity (defined as suffering from at least two chronic conditions) and polymedication (defined the prescription of at least five medications for their chronic conditions). Regarding, primary care services utilization, the total number of annual contacts was calculated, as well as the number of contacts according to type (health, administrative, laboratory), form of contact (face-to-face, telephone, home visit) and professional contact (family physician, nurse, social worker, midwife, physiotherapist and dentist). 

Sociodemographic, functional and clinical variables were extracted from Madrid’s primary care electronic medical record database on 30 June 2015, while primary care service utilization was recorded from 30 June 2015 to 30 June 2016. 

### 2.4. Statistical Analysis

To characterize the sample, descriptive statistics were calculated for all the variables in the study for the entire population with chronic diseases, as well as distinguishing between those suffering or not from multimorbidity, including the stratification of the multimorbid population by AMG risk level, sex and age group. Qualitative variables were described by counts and percentages, whereas quantitative variables were by means (standard deviation) or medians (interquartile range). Normality was measured with the Shapiro–Wilk test. For bivariate analysis, qualitative variables were compared via χ^2^ tests (or Fisher’s exact test when appropriate), and polytomous and quantitative variables with parametric or nonparametric tests. Multiple comparisons were adjusted by applying the Bonferroni correction. Factors associated with primary care utilization in the population with multimorbidity were identified by a multiple linear regression whose dependent variable was the total number of contacts with primary care and the independent variables those significantly associated with the total number of contacts with primary care determined in simple linear regression analyses. Results in bivariate and multivariable analyses were considered statistically significant if *p* < 0.05. Statistical analysis was performed with IBM SPSS Statistics version 25 software (IBM Corp., Armonk, NY, USA).

## 3. Results

In the healthcare area studied, 9866 (54.6%) patients had at least one chronic disease, and 6036 (33.3%) had multimorbidity. Patients with multimorbidity had a mean age of 64.1 years and 63.5% were women, while patients with only one chronic condition mean age was 42.6 and 58.1% were women. Regarding their functional status, 4.8% of patients with multimorbidity were immobilized, 2.4% institutionalized in a residence, 3.6% had a primary caregiver at home, 1.3% needed home support, and 0.7% received palliative care, these functional limitations being less frequent in patients with just one chronic disease. The mean number of chronic conditions in patients with multimorbidity was 3.5, and a total of 25.3% were polymedicated vs. 1.8% in chronic patients without multimorbidity and 64.2% were classified by AMG as low risk, 28.5% as medium risk and 7.3% as high-risk, whereas most chronic patients without multimorbidity were low-risk (Table 1).

According to the AMG stratification, mean age increased as the risk level did, 58.6 years for low risk, 72.8 years for medium and 78.2 years for high. High-risk patients registered the worst functional status and the highest complexity (30.3 in high-risk patients vs. 12.5 in medium and 5.2 in low-risk patients) and rates of polypharmacy (79.8% of the high-risk patients vs. 44.1% of the medium-risk and 10.8% of the low-risk patients) (Appendix A). 

Regarding sex, women presented a higher average age (65.2 years vs. 62.1 years), higher prevalence of immobilization (5.6% vs. 3.4%), institutionalization (2.8% vs. 1.7%), need for caregivers (4.1% vs. 2.9%) and higher mean number of chronic diseases (3.6 vs. 3.4) and proportion of polymedication (27.6% vs. 21.3%). Relative to age groups, patients over 65 years had a higher prevalence of women (67% vs. 59.8%), immobilization (9.1% vs. 0.3%), need for institutionalization (4.6% vs. 0.1%) and caregivers (6.9% vs. 0.3%) and a higher mean number of chronic conditions (4.1 vs. 2.8) as well as more polymedication (48.9% vs. 1.1%) (Appendix A).

The most prevalent chronic conditions within the population with multimorbidity were dyslipidemia (54.5%), hypertension (51.9%), obesity (22.9%), depression (17.7%) and osteoporosis (17.1%). In contrast, the most frequent diseases among the chronic population without multimorbidity were anxiety (17.7%), dyslipidemia (12.8%), asthma (11.6%), thyroid disorder (8.1%) and anemia (7.6%) (Table 2).

The most prevalent conditions among all risks were dyslipidemia and hypertension, followed by anxiety (30.0%) and thyroid disorder (21.2%) at the low risk level, obesity (26.9%) and diabetes (25.1%) at the medium risk level and dysrhythmias (43.6%) and neoplasia (37.3%) at the high risk level. (Appendix A). 

Some diseases were more prevalent among women, such as anxiety (31.6% vs. 20.7%), thyroid disorder (29.1% vs. 10.0%), osteoporosis (25.8% vs. 2%), depression (21.4% vs. 11.3%), anemia (12.2% vs. 6.8%), asthma (11.1% vs. 7.9%) and dementia (4% vs. 2.2%). In contrast, others were more prevalent among men: hypertension (57.1% vs. 48.9%), diabetes mellitus (21.9% vs. 13.6%), cirrhosis (9.3% vs. 6.0%), ischemic heart disease (10.1% vs. 3.5%), neoplasia (9.6% vs. 5.6%) and chronic obstructive pulmonary disease (COPD) (9.4% vs. 4.3%). Among patients aged >65 years, there was a higher prevalence of hypertension (72.1% vs. 31.1%), dyslipidemia (63.3% vs. 45.6%), osteoporosis (26.2% vs. 7.8%), diabetes (23.4% vs. 9.6%), neoplasia (9.5% vs. 4.5%), glaucoma (9.1% vs. 2.6%), dementia (6.3% and 0.2%) and stroke (6.8% vs. 1.5%), among other diseases. In contrast, alcohol abuse (8.1% vs. 3.6%), substance abuse (3.2% vs. 0.2%), anxiety (35.8% vs. 19.7%) and asthma (14.3% vs. 5.7%) were more prevalent in those ≤ 65 years of age. (Appendix A).

Regarding the use of primary care services, the mean number of annual contacts in patients with multimorbidity was 14.9 and 6.3 in patients with only one chronic disease. The mean of contacts rose with the level of risk (9.8 in low risk, 21.5 in medium and 34.1 in high), and was higher in women than men (15.3 vs. 14.3) and in those >65 years old than in younger patients (19.6 vs. 10.1). The preferred contact form was face-to-face and health-related. Concerning the professional contacted, primary care doctors received a mean of 7.8 visits and primary care nurses 4.4, while the physiotherapists, midwives, dentists and social workers were contacted less frequently (Table 3, Appendix A).

Sociodemographic and functional factors significantly associated with primary care utilization in patients with multimorbidity were age over 65 years (B Coefficient [BC] = 1.2; 95% Confidence Interval [CI]  =  0.3–2.0), female sex (BC = 1.04; 95% CI = 0.3–1.8) and having a primary caregiver (BC = 8.7; 95% CI  =  6.7–10.7). Clinical variables associated with greater utilization of primary care in patients with multimorbidity were complexity index (BC  =  0.5; 95% CI  =  0.4–0.6), risk level (BC  =  2.3; 95% CI  =  1.3–3.3), ≥ 3 chronic conditions (BC  =  1.2; 95% CI  =  0.4–2.0) and polymedication (BC = 5.1; 95% CI  =  4.0–6.1). The chronic conditions causing a higher use of primary care were dysrhythmia (BC = 5.4; 95% CI = 4.2–6.7), dementia (BC = 4.8; 95% CI = 2.8–6.8) and diabetes (BC = 2.3; 95% CI = 1.3–3.3) (Table 4).

## 4. Discussion

### 4.1. Main Findings

A total of 33.3% of the population from the healthcare area studied had multimorbidity. Their average age was almost 65 years old with a predominance of females, high complexity, medium and high risk levels, numerous chronic conditions and a greater need for assistance, care and polymedication compared to patients with only one chronic condition. The use of primary care services was notably high, mainly with family doctors, and was influenced by sociodemographic, functional and clinical factors.

### 4.2. Characteristics of the Population and Generalization of the Results

The prevalence of multimorbidity was 33.3%, close to the 37.2% described worldwide in a recent systematic review and meta-analysis [2], where regional estimates varied depending on the population, age group, and chronic conditions or multimorbidity definition considered [2,4]. As expected, the population with multimorbidity studied had an advanced mean age [12,13,14,15,16,17,18,19,20] and almost two-thirds were women, coinciding with the predominance of this sex within multimorbid populations described in other series from different regions around the world [13,14,17,18,19]. Patients with multimorbidity had higher levels of immobility, institutionalization and need for caregivers due to their functional impairment caused by their multiple chronic conditions and complexity, as observed in other populations with multimorbidity [16,20,21,22,23]. The most frequent chronic conditions were similar to other series of patients with multimorbidity across the globe [10,12,14,15,16,17,19,20,21,24], predominately cardiovascular, osteoarticular, psychiatric and neoplasms.

Our results coincided with the populations with multimorbidity described by Rizza et al. [13], Linden et al. [14] and Ibarra-Castillo et al. [19] identifying a higher prevalence of chronic diseases in women; the proportion of women decreased in the high risk level, but was still more representative than men. In line with a review focusing on the interplay between multimorbidity and functional impairment [23], high-risk patients presented more functional impairment and immobility and required more frequent caregivers and palliative care. Low-risk multimorbid patients often had two mild chronic conditions, while medium-risk patients normally had more than four diseases and were similar in frequencies and characteristics to pluripathological patients [16], whereas high-risk patients could be compared to complex chronic patients with more functional and fragile deterioration [12]. The most frequent diseases in high- and medium-risk patients were hypertension, diabetes, neoplasms, obesity and heart failure, while low-risk patients predominantly had dyslipidemia, anxiety and thyroid disorders, as observed in other adult populations with multimorbidity [12,13]. Polypharmacy was increased in high-risk patients, as a higher prescription of medication is frequently associated with higher complexity and greater comorbidities [16,25].

Women were older and had greater immobility, with an almost doubled need for care, more chronic diseases and more polypharmacy than men, as described previously [14,20,26]. Anxiety, thyroid disorder, osteoporosis, depression, anemia and dementia predominated in women, while hypertension, cirrhosis, COPD, ischemic heart disease and neoplasia predominated in men, in line with other studies analyzing populations with multimorbidity by sex [14,21,26].

Regarding age, patients with multimorbidity over 65 years had a female predominance, given that women have a longer lifespan [14,20,26]. Older patients presented with more immobilization, greater needs of care, and higher quantity and severity of chronic conditions, as observed by other authors [14,15,23,26]. Predominant chronic conditions in the elderly were hypertension, osteoporosis, diabetes, neoplasia, glaucoma, dementia and stroke, as described in other aged populations [15,21,26]. Almost half of the elderly were polymedicated, in contrast to 1% of younger patients, explained because multimorbidity, which increases with age, increases complications and health adverse events, so more medication is needed to control concomitant chronic diseases [25].

### 4.3. Use of Primary Healthcare Services

A remarkably high number of contacts with the primary care system was observed in the patients with multimorbidity, supporting the literature [10,12,17,18,24,26,27,28] and the fact that multimorbidity has been estimated to account for 78% of all primary care consultations [29]. Multimorbidity patients tripled the average primary care contacts registered by the general population of Madrid the previous year [30], being this threefold difference in accordance with data reported by other studies [17,24,27]. The family doctor was the most contacted type of professional, pursuant to the gatekeeping system of the Spanish National Health System [11] and with the mean annual contacts registered with this professional by Cassell et al. [17], Bähler et al. [18] and Soley-Bori et al. [27]. The number of contacts with a nurse was lower than expected according to Madrid’s model of care for addressing chronicity, which promotes a key role of nurses prioritizing most of the follow-up and care of the patients, while doctors should only intervene when medical care is needed or in situations of greater complexity [7]. Following this strategy, the role of the social worker was also expected to be higher. Regardless, telephone contacts and home visits should be increased in the context of care for multimorbidity patients [18,27], as these were truly low.

Every contact with primary care in multimorbidity patients increased according to the risk level [28]. The mean number of contacts with family doctors was higher among women, as they often report a worse perceived health status and have more minor affective disorders that can cause more doctor consultations [21]. In contrast, primary care nurse consultations were higher among men, perhaps due to less self-care capacity, as it is usually described among elderly men [31]. It is noteworthy how aged patients with multimorbidity patients had a nearly two-fold amount of total contact with primary care physicians compared to younger patients, since they usually have more complexity and comorbidities [18,24,28].

The variable with the greatest impact on the use of primary care services was having a primary caregiver, since patients who require this assistance often present with severe chronic diseases and functional and cognitive limitations and disabilities impeding them from getting healthcare services by themselves; thus, caregivers provide them assistance and facilitate access to healthcare services and transitions between healthcare professionals [32]. Being polymedicated was also highly associated with the utilization of healthcare services as polypharmacy is related to suffering a greater number of diseases with greater severity, mostly leading to higher use of services due to frequent dose adjustments by primary care professionals or due to errors in medication intake, adverse effects or interactions between drugs [25,27,28]. In the same way, primary care utilization was influenced by the AMG complexity degree, high risk level and having numerous chronic conditions, as observed in other studies with complex patients or with multimorbidity [12,18,26,28]. The association between advanced age in patients with multimorbidity and the use of services was evident [18,19,20,24,26,28]. As well, the female sex was significantly linked to higher healthcare utilization, probably because as women have a longer life expectancy they often suffer from a worse health status requiring more visits to the primary care system [13,14,26]; in addition, women typically evaluate their health and health-related outcomes worse than males [21]. Finally, the diseases dysrhythmia, diabetes and dementia were correlated with increased primary care utilization because these pathologies require continuous monitoring by primary care professionals, involving routine adjustment of medication dosage and frequent control of symptoms [33,34].

### 4.4. Limitations

The use of secondary data sources could cause possible biases of information linked to the variability in the way the different health professionals registered the diseases. Nevertheless, the use of clinical–administrative sources for epidemiological studies is widespread, providing real-world data and facilitating work with almost all individuals and not with partial samples, minimizing possible selection and memory biases. There could be patients not represented in the total population of the basic healthcare area if they had private insurance, but this is unlikely to bias the results because Madrid public health insurance covers almost the entire population [11].

Morbidity groupers have raised doubts about their transparency and their calculations of complexity, and commonly do not consider socioeconomic status, frailty or disability, need for care, clinical values or prognostic rating scales. AMG has overcome these problems and has proven its validity compared to other groupers [35]. Lastly, some chronic diseases may have not been considered by the AMG, which only takes into account the ones described by the Community of Madrid Strategy of Care for Patients with Chronic Diseases [7]; likewise, this problem affects any study with chronic diseases since no universal definition has been established.

Although the data were extracted from a unique healthcare area, all the results are representative of the Madrid population with chronic diseases and multimorbidity and could be extrapolated to the rest of the Spanish territory and outside Spain, since this healthcare area serves a widely heterogeneous group of people, including patients of different conditions and nationalities. Additionally, our results showed similar trends to the ones observed in other studies conducted around the world.

The COVID-19 pandemic introduced great challenges in the organization of health centers. Work systems and relationship models with patients may have experienced an increase in non-face-to-face contact. For that reason, the mean number of telephone contacts is likely to be higher nowadays, as this global event has shown that telephonic consultations are a useful tool to support the traditional care model. Nevertheless, despite that the data may seem outdated, the results obtained regarding the utilization of primary care services by the Madrid population described for the year of our study [36] are very close in numbers to those stated in the last Annual Report of Madrid Health Service [37]. Therefore, the healthcare utilization findings presented in this study depict the current reality and are of great importance and usefulness.

### 4.5. Implications

Patients with multimorbidity create a high care burden for the primary care system. These patients require the development of personalized management approaches made by primary care professionals, with the aim of improving the quality and continuity of care, as well as optimizing the healthcare services offered and reducing the costs derived from fragmented care [3,7]. For designing novel healthcare strategies, it is necessary to thoroughly characterize this population.

All the characteristics and use of primary care services in the multimorbid population described in this study provide valuable information that will help researchers, primary care professionals and healthcare policymakers. Understanding the findings depicted will facilitate the provision of holistic and evidence-based care and allocation of resources, focusing on the real situation and existing needs of patients with multimorbidity with the purpose of improving the quality and life expectancy of patients while also optimizing the utilization of primary care services, thereby reducing important healthcare costs and boosting the sustainability of healthcare systems [3,4,8]. Additionally, morbidity groupers such as the novel AMG support primary care professionals in identifying these patients and developing individualized interventions adapted to multimorbidity patient care needs, prioritizing those with higher risk and complexity [9,10], as recommended in the Madrid Care Strategy for people with chronic diseases [7].

## 5. Conclusions

Multimorbidity patients represent an important percentage of the population. They are older, with a female predominance and important needs of care, suffering from several comorbidities and polypharmacy, a situation that worsens with the AMG’s level of risk and complexity. The utilization of primary care services is extremely high, mostly directed to family doctors and mainly associated with sociodemographic factors such as old age and female sex, functional factors such as having a primary caregiver, and clinical factors such as complexity index, high risk level, the presence of numerous chronic diseases and polymedication. This study provides novel data on risk levels by the novel AMG grouper in multimorbidity patients, as well as characterizes the utilization of health services and needs of care. Understanding these valuable real-world data could profoundly help primary care professionals and policymakers to improve coordination between healthcare professionals and allocation of services to optimize multimorbidity management and ensure a better quality of life for patients as well as to reduce costs derived from their extensive health service utilization.

## Figures and Tables

**Table 1 healthcare-12-00270-t001:** Sociodemographic, functional and clinical characteristics of the total population with chronic conditions and with or without multimorbidity.

Chronic Patients	Total	No Multimorbidity		Multimorbidity		*p*-Value
*n* (%)	9866 (100)	3830 (38.8)	95% CI	6036 (61.2)	95% CI	
**Sociodemographic variables**						
Female	6056 (61.4)	2225 (58.1)	56.5–59.6	3831 (63.5)	62.2–64.7	<0.001
Age *	55.7 (20.8)	42.6 (18.5)	42.1–43.2	64.1 (17.6)	63.6–64.5	<0.001
≤65 years	6383 (64.7)	3408 (89.0)	88.0–90.0	2975 (49.3)	48.0–50.5	<0.001
>65 years	3483 (35.3)	422 (11.0)	10.0–12.0	3061 (50.7)	49.4–51.9	
Origin Spain	8078 (81.9)	3026 (79.0)	77.7–80.3	5052 (83.7)	82.8–84.6	<0.001
Europe	367 (3.7)	178 (4.6)	3.9–5.3	189 (3.1)	2.7–3.6	
Rest of the world	1421 (14.4)	626 (16.3)	15.2–17.5	795 (13.2)	12.3–14.9	
**Functional variables**						
Immobilized	300 (3.0)	12 (0.3)	0.1–0.5	288 (4.8)	4.2–5.3	<0.001
Institutionalized	161 (1.6)	16 (0.4)	0.2–0.6	145 (2.4)	2.0–2.8	<0.001
Primary caregiver	229 (2.3)	9 (0.2)	0.08–0.4	220 (3.6)	3.2–4.1	<0.001
Home support	80 (0.8)	3 (0.1)	0.07–0.2	77 (1.3)	0.09–1.6	<0.001
Palliative care	44 (0.4)	3 (0.1)	0.03–0.2	41 (0.7)	0.04–0.09	<0.001
**Clinical variables**						
High Risk Level	444 (4.5)	4 (0.1)	0.01–0.2	440 (7.3)	6.6–7.9	<0.001
Medium	1784 (18.1)	63 (1.6)	1.2–2.1	1721 (28.5)	27–30	
Low	7638 (77.4)	3763 (98.3)	97.8–98.6	3875 (64.2)	63–65	
Complexity weight *	6.7 (7.0)	2.9 (2.7)	2.8–3.0	9.1 (7.8)	8.9–9.3	<0.001
Chronic conditions *	2.5 (1.8)	1 (0.5)	0.8–1.2	3.5 (1.7)	3.4–3.6	<0.001
Polymedicated	1598 (16.2)	70 (1.8)	1.4–2.3	1528 (25.3)	24.2–26.4	<0.001

* Mean (standard deviation). CI: Confidence interval.

**Table 2 healthcare-12-00270-t002:** Comorbidities in the total population with chronic conditions and with or without multimorbidity.

Comorbidities		Total	No Multimorbidity		Multimorbidity		*p*-Value
*n* (%)		9866 (100)	3830 (38.8)	95% CI	6036 (61.2)	95% CI	
**Haematic** **comorbidity**	Anemia	908 (9.2)	290 (7.6)	8.6–9.8	618 (10.2)	9.5–11.0	<0.001
	HIV	55 (0.6)	19 (0.5)	0.4–7.0	36 (0.6)	0.4–0.8	0.514
**Digestive** **comorbidity**	Cirrhosis	479 (4.9)	45 (1.2)	4.4–5.3	434 (7.2)	6.5–7.8	<0.001
	Inflammatory bowel disease	75 (0.8)	24 (0.6)	0.6–0.9	51 (0.8)	0.6–1.1	0.224
	Gastric ulcer	175 (1.8)	21 (0.5)	1.5–2.0	154 (2.6)	2.1–2.9	<0.001
	Chronic pancreatitis	8 (0.1)	1 (0.01)	0.01–0.2	7 (0.1)	0.03–0.2	0.162
	Cystic fibrosis	3 (0.03)	0 (0)	0.001–0.006	3 (0.01)	0.001–0.1	0.287
**Ocular comorbidity**	Glaucoma	395 (4.0)	39 (1.0)	3.6–4.4	356 (5.9)	5.3–6.45	<0.001
**Cardio** **vascular comorbidity**	Hypertension	3418 (34.6)	285 (7.4)	33.7–35.6	3133 (51.9)	50.6–53.2	<0.001
	Dysrhythmias	696 (7.1)	46 (1.2)	6.5–7.6	650 (10.8)	10.0–11.5	<0.001
	Chronic heart failure	240 (2.4)	2 (0.1)	2.1–2.7	238 (3.9)	3.4–4.4	<0.001
	Coronary disease	370 (3.8)	11 (0.3)	3.4–4.1	359 (5.9)	5.3–6.5	<0.001
	Valvular heart disease	196 (2.0)	11 (0.3)	1.7–2.3	185 (3.1)	2.4–3.5	<0.001
**Musculo** **skeletal comorbidity**	Osteoarthritis	1055 (10.7)	92 (2.4)	10.1–11.3	963 (16.0)	15.0–16.9	<0.001
	Osteoporosis	1113 (11.3)	79 (2.1)	10.6–11.9	1034 (17.1)	16.2–18.1	<0.001
	Arthritis	235 (2.4)	47 (1.2)	2.1–2.7	188 (3.1)	2.7–3.5	<0.001
	Lupus	5 (0.1)	1 (0.01)	0.06–0.9	4 (0.1)	0.001–0.013	0.655
	Vasculitis	27 (0.3)	1 (0.01)	0.1–0.5	26 (0.4)	0.3–0.6	<0.001
**Neurological** **comorbidity**	Dementia	213 (2.2)	13 (0.3)	1.8–2.5	200 (3.3)	2.9–3.8	<0.001
	Stroke	267 (2.7)	15 (0.4)	2.4–3.0	252 (4.2)	3.7–4.7	<0.001
	Parkinson	85 (0.9)	1 (0.01)	0.7–1.1	84 (1.4)	1.1–1.7	<0.001
	Epilepsy	187 (1.9)	60 (1.6)	1.6–2.2	127 (2.1)	1.7–2.5	0.056
	Multiple sclerosis	32 (0.3)	14 (0.4)	0.2–0.4	18 (0.3)	0.2–0.5	0.567
**Psychiatric** **comorbidity**	Alcohol abuse	407 (4.1)	55 (1.4)	3.7–4.5	352 (5.8)	5.2–6.4	<0.001
	Substance abuse	130 (1.3)	30 (0.8)	1.1–1.5	100 (1.7)	1.3–1.9	<0.001
	Anxiety	2345 (23.8)	678 (17.7)	22.9–24.6	1667 (27.6)	2.6–2.9	<0.001
	Depression	1251 (12.7)	181 (4.7)	12.0–13.3	1070 (17.7)	16.8–18.7	<0.001
	Bipolar illness	66 (0.7)	10 (0.3)	0.5–0.8	56 (0.9)	0.7–1.2	<0.001
	Psychotic disorder	74 (0.8)	9 (0.2)	0.6–0.9	65 (1.1)	0.8–1.3	<0.001
**Respiratory** **comorbidity**	COPD	389 (3.9)	22 (0.6)	3.5–4.3	367 (6.1)	5.5–6.7	<0.001
	Asthma	1044 (10.6)	444 (11.6)	10.0–11.2	600 (9.9)	9.2–10.7	0.009
**Endocrine** **comorbidity**	Dyslipidemia	3780 (38.3)	491 (12.8)	37.3–39.3	3289 (54.5)	53.2–55.7	<0.001
	Diabetes mellitus	1063 (10.8)	60 (1.6)	10.2–11.4	1003 (16.6)	15.7–17.6	<0.001
	Obesity	1625 (16.5)	241 (6.3)	15.7–17.2	1385 (22.9)	21.9–24.0	<0.001
	Thyroid disorder	1646 (16.7)	310 (8.1)	15.9–17.4	1336 (22.1)	21.1–23.2	<0.001
**Renal comorbidity**	Renal chronic failure	142 (1.4)	2 (0.1)	1.0–1.7	140 (2.3)	1.9–2.7	<0.001
	Repeat urinary tract infection	497 (5.0)	150 (3.9)	4.6–5.5	347 (5.7)	5.2–6.3	<0.001
**Cancer** **comorbidity**	Any cancer	481 (4.9)	54 (1.4)	4.4–5.3	427 (7.1)	76.4–7.7	<0.001
	Breast	74 (0.8)	12 (0.3)	0.5–0.9	62 (1.0)	0.8–1.3	<0.001
	Prostate	66 (0.7)	5 (0.1)	0.5–0.8	61 (1.0)	0.8–1.3	<0.001
	Skin	60 (0.6)	8 (0.2)	0.4–0.8	53 (0.9)	0.6–1.1	<0.001
	Colorectal	57 (0.6)	4 (0.1)	0.4–0.8	52 (0.9)	0.6–1.1	<0.001
	Bladder	35 (0.4)	1 (0.01)	0.2–0.6	34 (0.6)	0.4–0.7	<0.001
	Lung	34 (0.3)	1 (0.01)	0.2–0.4	33 (0.5)	0.3–0.7	<0.001
	Cervix	18 (0.2)	3 (0.1)	0.1–0.3	12 (0.2)	0.1–0.4	0.054
	Liver	7 (0.1)	2 (0.1)	0.01–0.2	5 (0.1)	0.01–0.2	0.578
	Gastric	8 (0.1)	0 (0)	0.01–0.2	8 (0.1)	0.04–0.2	0.024
	Pancreas	6 (0.1)	0 (0)	0.01–0.2	6 (0.1)	0.02–0.2	0.051
	Renal	12 (0.1)	0 (0)	0.01–0.2	12 (0.2)	0.1–0.3	0.006
	Endometrium	5 (0.1)	0 (0)	0.01–0.2	5 (0.1)	0.01–0.2	0.075
	Leukemia	27 (0.3)	2 (0.1)	0.2–0.4	25 (0.4)	0.2–0.6	<0.001
	Lymphoma	48 (0.5)	1 (0.01)	0.3–0.7	47 (0.8)	0.6–1.0	<0.001

CI: Confidence interval. COPD: Chronic obstructive pulmonary disease.

**Table 3 healthcare-12-00270-t003:** Annual primary care service utilization of the total population with chronic conditions and with or without multimorbidity.

Primary Care Contacts	Total	No Multimorbidity		Multimorbidity		*p*-Value
Mean (SD)	9866 (100%)	3830 (38.8%)	95% CI	6036 (61.2%)	95% CI	
**Total Annual Contacts**	11.5 (14.6)	6.3 (9.2)	6.0–6.6	14.9 (16.4)	14.5–15.3	<0.001
**Type of contact**						
Health-related	9.9 (12.8)	5.5 (8.0)	5.2–5.7	12.8 (14.3)	12.4–13.2	<0.001
Administrative	0.8 (3.3)	0.4 (1.9)	0.3–0.5	1.2 (3.9)	1.1–1.3	<0.001
Laboratory	0.7 (1.3)	0.4 (0.9)	0.3–0.5	0.9 (1.4)	0.9–1.0	<0.001
**Form of contact**						
Face-to-face	10.6 (12.5)	6.1 (8.5)	5.8–6.4	13.44 (13.8)	13.1–13.8	<0.001
Telephone	0.4 (2.2)	0.1 (0.9)	0.08–0.14	0.5 (2.7)	0.5–0.6	<0.001
Home visit	0.6 (3.9)	0.07 (1.0)	0.04–0.1	0.9 (4.9)	0.8–1.1	<0.001
**Professional contacted**						
Doctor	6.0 (7.1)	3.3 (4.8)	3.2–3.5	7.8 (7.7)	7.6–8.0	<0.001
Nurse	3.1 (7.0)	1.2 (3.7)	1.1–1.3	4.4 (8.2)	4.2–4.6	<0.001
Physiotherapist	0.3 (2.0)	0.2 (1.6)	0.1–0.3	0.3 (2.2)	0.3–0.4	0.064
Midwife	0.1 (7.2)	0.2 (1.0)	0.1–0.3	0.07 (0.4)	0.06–0.08	<0.001
Dentist	0.05 (0.4)	0.05 (0.4)	0.05–0.08	0.05 (0.4)	0.04–0.06	0.010
Social worker	0.07 (0.6)	0.02 (0.2)	0.01–0.03	0.1 (0.7)	0.09–0.12	<0.001

SD: Standard Deviation. CI: Confidence interval.

**Table 4 healthcare-12-00270-t004:** Factors associated with the use of primary healthcare services in patients with multimorbidity.

Variables	B Coefficient	95% CI	*p*-Value
Primary caregiver	8.70	6.72–10.69	<0.001
Dysrhythmia	5.43	4.20–6.65	<0.001
Polymedicated	5.05	4.00–6.10	<0.001
Dementia	4.83	2.83–6.84	<0.001
Risk level	2.29	1.26–3.32	<0.001
Diabetes mellitus	2.27	1.28–3.27	<0.001
≥3 chronic diseases	1.20	0.37–2.04	0.005
Age > 65 years	1.15	0.30–2.01	0.008
Female sex	1.04	0.30–1.78	0.006
Complexity weight *	0.46	0.38–0.55	<0.001

Backward stepwise regression, R^2^ = 0.301. * Continuous variable. CI: Confidence Interval.

## Data Availability

The datasets generated and analyzed during the current study are not publicly available because they belong to the Madrid Health Service, but they are available from the corresponding author on reasonable request.

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
