# Peer review of "Utilization of Primary Healthcare Services in Patients with Multimorbidity According to Their Risk Level by Adjusted Morbidity Groups: A Cross-Sectional Study in Chamartín District (Madrid)"

_healthcare, 2024, doi:10.3390/healthcare12020270_

Round 1

Reviewer 1 Report

Comments and Suggestions for Authors

This study could be of great interest for the planning of Primary Health Care (PHC) in Madrid if it had used updated data.

The COVID-19 pandemic introduced great challenges in the organization of health centers. Work systems were modified and relationship models with patients were altered, notably increasing non-face-to-face contacts.

But this work presents very old data. Eight years old. It would lead to confusion because it presents a reality that is already past.

I suggest the authors repeat the work with updated data and including in their analysis the demand in emergency services and the situation of waiting lists in PHC.

Reviewer 2 Report

Comments and Suggestions for Authors

The reviewed manuscript covers very sound topic - primary health care for patients with multimorbidity. The multimorbidity is very actual not only as clinical problem but also as a research  problem.  The manuscript is well-written. I have no remarks to the design of the study and statistical methods of the data analysis. My only suggestion for authors of the manuscript is add short description in the Materials and Methods section of the manuscript how patients were grouped into the Adjusted Morbidity Groups (AMG) risk levels because it is not presented in details and just references No 7 and 9 in the list of references indicated. But these two articles are published in Spanish therefore the methods of classification of the patients into AMG risk level groups can be easily understood only for some part of readers.

Reviewer 3 Report

Comments and Suggestions for Authors

Thank you for the opportunity to review the manuscript. The manuscript requires major revision, with a focus on strengthening the introduction section, methods, and discussion to improve the clarity and coherence of the text. Authors are encouraged to review and implement these suggestions according to the specific details of their study:

Abstract:

-        The abstract needs to be reorganized to present background information, methods employed, results obtained, and the study's conclusion. Notably, the abstract should also include projected results such as the B Coefficient and 95% CI.

Introduction Section:

-        The second paragraph's initial statement is cited for validation.

-        The third paragraph's first sentence is clarified to specify what "these" refers to.

-        A crucial aspect of the overall introduction section is addressing the lack of an introduction to previous literature and a rationale for the current research. The authors must fill this research gap to enhance the comprehensibility of the introduction.

Materials and Methods:

-        The materials and methods section is lacking organization, and key details about data collection, data collectors, and national representativeness are not clearly presented.

-        A separate section for "variables" is recommended, providing a clear explanation of how variables, including multimorbidity, were coded. Proper explanation of variable coding is essential to avoid confusion among readers and reviewers.

-        Due to the unclear mention of variables in the method section, understanding the results becomes challenging for the reader and reviewer.

Discussion:

-        The discussion section mentions "other multimorbid populations" without specifying the studies. The authors are advised to mention and cite the exact studies for clarity.

-        The term "other studies" is used without clarification, and it is suggested to provide more details for better understanding.

-        The implication section needs strengthening, elaborating on how the study contributes to the enhancement of the healthcare system for patients with multimorbidity. Clear explanations regarding the implications are necessary.

Reviewer 4 Report

Comments and Suggestions for Authors

I read the manuscript and it is well written but I have few concerns:

1.       Elaborate further on the application of AMG and its significance in this study, incorporating AMG into the study title as a crucial keyword for patient risk classification.

2.       Ensure that the title specifies the data's origin from a single center, the Chamartín district. Provide a description of the geographical location and healthcare system within the study area.

3.       Justify the use of retrospective data in June 2015, considering the potential limitations due to a gap of more than five years. Acknowledge the possibility of changes and improvements in healthcare utilization during this period. How far the findings could be applied in the current year

4.       Define qualitative variables and explain on the study criteria, detailing the process of sample selection.

5.       In Table 2, explain the rationale behind categorizing HIV under Haematic comorbidities and provide justification for this classification.

6.       Provide a comprehensive description of the dependent and independent variables used in the linear regression analysis to identify factors influencing healthcare services utilization. Present detailed results from both simple and multiple linear regression analyses.

7.       The discussion section will be revisited for review following the adjustment of the linear regression input.

Round 2

Reviewer 1 Report

Comments and Suggestions for Authors

I have read the authors' response. I do not agree.

I do not feel right allowing this article to be published.

Eight years since data collection. The objectives of the study were affected by the covid pandemic.

Reviewer 3 Report

Comments and Suggestions for Authors

Now the paper is much improved and can be acceptable.

Author Response

We want to thank Reviewer 3 for the time and effort dedicated to the review process and we really appreciate that he/she thinks that our manuscript is now much improved and can be acceptable for publication. His/her valuable comments in the first revision greatly contributed to the improvement of our manuscript. 

Reviewer 4 Report

Comments and Suggestions for Authors

I looked over the revised manuscript, and it's significantly better now. The content has been enhanced a lot. Thanks for considering all the improvement points.

Author Response

We want to thank Reviewer 4 for the time and effort dedicated to the review process. We really appreciate that he/she thinks that our manuscript is significantly better now and that the content has been enhanced a lot. His/her valuable comments in the first revision greatly contributed to the improvement of our manuscript.